# Nutritional Characterization of *Strychnos madagascariensis* Fruit Flour Produced by Mozambican Communities and Evaluation of Its Contribution to Nutrient Adequacy

**DOI:** 10.3390/foods11040616

**Published:** 2022-02-21

**Authors:** Sandra S. I. Chemane, Mafalda Ribeiro, Edgar Pinto, Susana C. M. Pinho, Zita Sá Martins, Agostinho Almeida, Isabel M. P. L. V. O. Ferreira, Maida Khan, Olívia Pinho, Susana Casal, Olga Viegas

**Affiliations:** 1Faculdade de Ciências da Nutrição e Alimentação da Universidade do Porto, 4150-180 Porto, Portugal; s.chemane.se@gmail.com (S.S.I.C.); olgaviegas@fcna.up.pt (O.V.); 2LAQV/REQUIMTE, Laboratório de Bromatologia e Hidrologia, Departamento de Ciências Químicas, Faculdade de Farmácia da Universidade do Porto, 4050-313 Porto, Portugal; mribeiro@ff.up.pt (M.R.); ecp@ess.ipp.pt (E.P.); susanapinho7@gmail.com (S.C.M.P.); zmartins@ff.up.pt (Z.S.M.); isabel.ferreira@ff.up.pt (I.M.P.L.V.O.F.); sucasal@ff.up.pt (S.C.); 3Departamento de Engenharia Rural, Faculdade de Agronomia e Engenharia Florestal, Universidade Eduardo Mondlane, Maputo 257, Mozambique; 4Department of Environmental Health, School of Health, P. Porto, 4200-072 Porto, Portugal; 5LAQV/REQUIMTE, Laboratório de Química Aplicada, Departamento de Ciências Químicas, Faculdade de Farmácia da Universidade do Porto, 4050-313 Porto, Portugal; aalmeida@ff.up.pt; 6Departamento de Engenharia Química, Faculdade de Engenharia, Universidade Eduardo Mondlane, Maputo 257, Mozambique; maida.khan@uem.ac.mz

**Keywords:** monkey orange, fruit flour, macronutrients, micronutrients, indigenous fruits, estimated daily intake

## Abstract

The indigenous fruit *Strychnos madagascariensis* is usually processed to flour, called *nfuma*, being highly consumed during staple food shortage. This study aimed to evaluate the nutritional composition of *nfuma* and its nutrient adequacy. Flours from four districts of Mozambique were analyzed using AOAC methods for proximate composition, HPLC for sugar, amino acids (AA), vitamin E and carotenoids and ICP-MS and FAAS for minerals. The results showed that *nfuma* stands out for its high content of fat (26.3–27.8%), mainly oleic acid, fiber (>6%), vitamin E (6.7 to 8.0 mg/100 g) and carotenes (2.2 to 2.6 mg/100 g). The main amino acids of *nfuma* protein were Arg, Asp and Glu, and Lys was the limiting one. The mineral composition reveals K (~1200 to 1700 mg/100 g) as the main macromineral followed by Mg > Ca > Na. The main trace element was Mn (~4 mg/100 g) followed by Fe > Zn > Cu > Cr > Co. Aluminum (~3 mg/100 g) was the main non-essential element and Rb, Ni, Sr, Ba, V, Cd were also quantified. Assuming the daily consumption of 50 g, *nfuma* provides 82% of Vitamin A dietary reference value for toddlers, while the consumption of 100 g contributes to 132% and 60% of Mn and vitamin A DRV for adults, respectively. Despite the nutritional advantages of *nfuma*, this flour can be a source of Ni, highlighting the importance of the study of good practices in its preparation to decrease the exposure to non-essential elements.

## 1. Introduction

In recent years, the increase in knowledge about the protective role of fruits and vegetables has led to an increase in campaigns to promote their consumption for better health. Although fruits and vegetables can be consumed in *natura*, since fresh products are highly perishable, they can be processed to increase their shelf life and maintain (or even improve) their nutritional quality and sensory characteristics [1].

Indigenous fruits have been receiving considerable attention from the scientific community as they can be important contributors to the diet of people in developing countries, reducing nutritional deficiencies and food insecurity, as well as improving the health and economic status of those populations [2,3]. In addition, they can be exploited by the agro-industry and become a source of income for local communities in the future [4,5]. Indigenous fruits are easily accessible to the most vulnerable people because fruit trees are not farmed and often grow in forests and around homes and fields. However, in African countries, indigenous fruits are still underutilized, while several communities are food insecure and, consequently, malnourished [5,6,7,8]. 

*Strychnos* spp. (monkey orange) is an indigenous fruit tree known for its edible fruits and drought tolerance. However, it has been labeled as a “lost fruit”—fruit that have potential as food- and cash-crops, but little attention has been paid by scientists, policymakers, and the world at large [8]. Thus, little attention has been paid to its potential commercialization, due to limited knowledge and disseminated information compared to many other exotic fruits [9]. Despite some imprecision in species differentiation, five species are prevalent and most consumed in southern Africa (*S. innocua*, *S. cocculoides*, *S. pungens S. spinosa*, and *S. madagascariensis*). Due to their seasonality and high perishability, traditional fruit processing is a very common practice, in addition to their immediate consumption as fresh fruits, with *S. madagascariensis* and *S. innocua* being processed preferably into dry products [10]. Significant variations in the nutritional composition of these fruits have been reported in the literature, and scarce information is available concerning their processed products [9].

Mozambique has numerous native and exotic fruit species that are important for rural communities’ survival in times of food shortages. The *S. madagascariensis* fruit, known in southern Mozambique as “macuácua”, can be consumed immediately in natura, but it is usually processed into flour by local communities to increase the stability and shelf life of the fruit. Once harvested, the fruit pulp is first dried under the sun and roasted over a fire, and then ground to produce *nfuma* flour, which is consumed by local communities as a snack or as a complement of staple foods in times of food scarcity. However, there is practically no data on its nutritional value. Thus, this study aimed to evaluate the nutritional composition of *nfuma*, the fruit flour of *S. madagascariensis,* and its adequacy in terms of nutrients in light of current recommendations.

## 2. Materials and Methods

### 2.1. Fruit Collection and Flour Preparation

Fruit samples were collected by residents of four different districts (Marracuene, Manhiça, Chókwè and Chicualacuala) in southern Mozambique. A total of 480 fresh fruits, weighing 510 ± 75 g (*n* = 12 fruits, randomly selected), were harvested per region (eight randomly selected trees; 15 fruits per tree; three different times), during summer, from October to December. *Nfuma* was traditionally prepared by local residents. Briefly, 120 ripe fruits were broken and the orange pulp with seeds was left to dry in the sun (2 to 4 days), to facilitate the removal of the seeds. Then, the pulp was roasted at about 50–60 °C (temperature measured using a digital thermocouple with a surface probe), on a metal plate under fire, for about 1 h. The dried pulp was then ground into flour using a pestle and mortar, producing about 5 kg of flour. The process was repeated three independent times per region. A schematic description of the sampling methodology is provided in Figure 1.

### 2.2. Chemical Analysis

#### 2.2.1. Proximate Analysis

Moisture, crude fat, protein and ash contents were determined according to the Association of Official Analytical Chemists (AOAC) methods 925.09, 920.39, 992.15 and 923.03, respectively [11]. Total dietary fiber was determined by the enzymatic-gravimetric method based on American Association of Cereal Chemists 32–05.01 method and AOAC 985.29 method, according to Martins et al. [12]. Carbohydrates were calculated by the differential method and the energy value was determined based on Regulation (EU) No 1169/2011 [13]. That is, the energy content was calculated from the amount of protein, fat, available carbohydrates and fiber using the factors 17, 37, 17 and 8 kJ per gram (4, 9, 4 and 2 kcal per gram), respectively.

#### 2.2.2. Low Molecular Weight Carbohydrates Determination by HPLC-RI

The low molecular weight carbohydrates (mono and disaccharides) were determined by high-performance liquid chromatography with refractive index detection (HPLC-RI), as described by Santos et al. [14], with some modifications. Briefly, five hundred milligrams of flour were accurately weighed into a centrifuge tube. Prior to sugar extraction, the oil was removed from the flour with the aid of three 5 mL portions of hexane, discarded after centrifugation, and the solid residue was left under a nitrogen stream to evaporate the solvent. The defatted flour was then mixed with 5 mL of ethanol (50% *v*/*v*), for sugar extraction. The suspension was stirred for 30 s and the extraction was carried out in an ultrasonic bath (FungiLab, Barcelona, Spain) for 30 min at 50 °C. Thereafter, the mixture was centrifuged at 5000× *g* at 4 °C for 10 min and 2 mL of the supernatant were left under a nitrogen stream to reduce the ethanol fraction. The final volume was then rigorously adjusted to 2 mL with acetonitrile and allowed to stand for 20 min. The final solution was centrifuged at 5000× *g* for 10 min at 4 °C and filtered through a 0.22 μm PTFE filter prior to injection.

#### 2.2.3. Fatty Acids Composition by GC-FID

The fatty acid composition of extractable lipids was evaluated as methyl esters derivatives by gas chromatography with flame ionization detection (GC-FID), using alkaline trans-esterification with methanolic potassium hydroxide, as detailed in Regulation EEC 2568/91 [15]. The analysis was performed using a Chrompack CP 9001 gas chromatograph (Middelburg, the Netherlands), equipped with a split–splitless injector, a flame ionization detector, an autosampler (Chrompack CP 9050) and a 50 m × 0.25 mm id fused silica capillary column coated with SelectFAME. Helium was used as carrier gas at an internal pressure of 120 kPa. The detector and injector temperatures were 250 and 230 °C, respectively. The results were initially expressed as a relative percentage of each fatty acid methyl ester, without discriminating positional and geometric isomers, calculated by internal normalization of the chromatographic peak areas after standardization of the detector response with the certified reference standard, and converted to the flour mass based on the determined fat content.

##### Atherogenicity and Thrombogenicity Indexes

The nutritional quality parameters atherogenic index (AI) and thrombogenic index (TI) were calculated according to [16]:(1)$$\mathrm{AI}=\frac{C12:0+4\;\times \;C14:0+C16:0}{\mathrm{MUFA}+n3\;\mathrm{PUFA}+n6\;\mathrm{PUFA}}$$
(2)$$\mathrm{TI}=\frac{C14:0+C16:0+C18:0}{0.5\;\times \;\mathrm{MUFA}+0.5\;\times \;n6\;\mathrm{PUFA}+3\;\times \;n3\;\mathrm{PUFA}+\frac{n3\;\mathrm{PUFA}}{n6\;\mathrm{PUFA}}}\;$$
where MUFA and PUFA correspond to monounsaturated and polyunsaturated fatty acids, respectively.

#### 2.2.4. Analysis of Vitamin E and Carotenoids by HPLC-DAD/FLD

The vitamin E and total carotene contents of the lipid extract were determined by normal-phase high-performance liquid chromatography with diode-array and fluorescence detection (HPLC-DAD/FLD), as described in [17]. An exact amount of fat was dissolved in hexane, an appropriate volume of the internal standard solution (tocol) was added, and the mixture was homogenized by stirring. A normal phase silica column (Supelcosil TM LC-SI; 7.5 cm × 3 mm; 3 µm) (Supelco, Bellefonte, PA, USA), conditioned at 25 °C and eluted with a gradient of 1,4-dioxane in hexane at a flow rate of 0.75 mL/min was used. Detection was programmed for excitation at 290 nm and emission at 330 nm for tocols and 450 nm for carotenes. The different vitamin E compounds were identified by comparing retention times with standards, and quantified through individual calibration curves, being expressed in mg of tocopherol/100 g of flour. β-carotene was also quantified based on a calibration curve.

#### 2.2.5. Amino Acid Composition by HPLC-FLD and Assessment of Protein Quality

Amino acids (Asp, Glu, Ser, Hi, Gly, Thr, Arg, Ala, Tyr, Val, Met, Phe, Ile, Leu, Lys, Pro, Trp, Cys, where Asp means aspartic acid/asparagine and Glu means glutamic acid/glutamine) were analyzed by HPLC-FLD, after hydrolysis and derivatization with 9-fluorenylmethyl chloroformate and O-phthaldialdehyde, according to Benhammouche et al. (2021). Briefly, about 100 mg of flour (±3.5 mg of protein) was weighed into a glass crimp vial. An amount of 10 mL of hydrochloric acid solution HCl 6 M containing 0.5% (w/v) phenol were added, sealed and the acid hydrolysis was performed at 110 °C for 24 h. An amount of 1 mL from the resulting hydrolysate was taken and neutralized with NaOH 6 N, and the final volume was made up to 10 mL with borate buffer (0.1 M). An amount of 32 µL of the neutralized solution was mixed with 8 µL of internal standard 250 µM (Norvaline) and 40 µL O-phthaldialdehyde and 20 µL 9-fluorenylmethyl chloroformate were added. Trp was determined separately using alkaline hydrolysis (NaOH 4.2 N, for 18 h at 110 °C). The resulting derivatization products were then subjected to HPLC analysis under the conditions detailed in [18]. Amino acids (AAs) content was reported as mg of AA/g protein.

In order to evaluate the quality of protein in *nfuma*, the essential amino acid profile (EAA) scores (EAAS) and EAA index (EAAI) were calculated using the following equations [19]:(3)$$\mathrm{EAAS}=\frac{{\mathrm{EAA}}_{\mathrm{test}\;\mathrm{protein}\;(\mathrm{mg}/g)}}{{\mathrm{EAA}}_{\mathrm{reference}\;\mathrm{protein}\;(\mathrm{mg}/g)}}$$
(4)$$\mathrm{EAAI}=\sqrt[{n}]{\mathrm{EAAS}\;1\times \mathrm{EAAS}\;2\times \mathrm{EAAS}\;3\times \mathrm{EAAS}\;n\;}\times \;100\;$$
where n is the number of amino acids included in the calculation. The reference protein used was the FAO/WHO EAAS pattern from the joint [20].

#### 2.2.6. Mineral Analysis

Mineral analysis was performed according to Pinto et al. [21]. Sample mineralization was performed by microwave-assisted closed-vessel acid digestion using an MLS-1200 Mega high-performance microwave digestion unit (Milestone, Sorisole, Italy) equipped with an HPR-1000/10 S rotor. Microminerals determination was performed by inductively coupled plasma-mass spectrometry (ICP-MS) using an iCAP™ Q instrument (Thermo Fisher Scientific, Bremen, Germany) and measuring the following elemental isotopes (m/z ratios): ^7^Li, ^9^Be, ^27^Al, ^51^V, ^52^Cr, ^55^Mn, ^59^Co, ^60^Ni, ^65^Cu, ^66^Zn, ^75^As, ^82^Se, ^85^Rb, ^88^Sr, ^114^Cd, ^133^Cs, ^137^Ba, ^205^Tl, ^208^Pb and ^209^Bi. The determination of Ca, Mg, Fe, Na and K was performed by flame atomic absorption spectroscopy (FAAS) using a PerkinElmer (Überlingen, Germany) AAnalyst 200 instrument. Limits of detection (LOD) and limits of quantification (LOQ) were estimated from the analysis of 10 digestion blanks. The results are presented in Appendix A (Appendix A). For quality control purposes, the certified reference material (CRM) IRMM 807 (rice flour, supplied by EC Institute for Reference Materials and Measurements, Geel, Belgium) and BCR 679 (white cabbage, supplied by EC Institute for Reference Materials and Measurements, Geel, Belgium) were analyzed under the same conditions as the samples. The results obtained were in good agreement with the certified values (recoveries ranging from 95.7% to 108.6%), proving that the method accuracy was adequate.

#### 2.2.7. Estimated Intake of Nutrients and Non-Essential Elements

##### Evaluation of Nutrient Adequacy 

The estimated daily intake (EDI) of nutrients and energy, as % of EFSA dietary reference values (DRVs) [22], was calculated assuming an average daily consumption of 100 g of *nfuma*. DRVs include a set of nutrient reference values: population reference intakes (PRIs), average requirements (ARs), adequate intakes (AIs) and reference intake (RIs). For nutrients that have both PRI and AR, the EDI was calculated based on PRI, since it corresponds to the intake level that meets the needs of all people in a population [22]. Regarding energy, the AR for adults with a physical activity level (PAL) between 1.4 and 2.0 was calculated as the mean value of AR for all age groups between 18 and 79 years. The AI and AR (g/day) for macronutrients were calculated based on the lower and upper limit of AR range for energy.

##### Estimated Intake of Non-Essential Elements

The estimated intake (EI) was calculated based on the elemental content (C_element_: µg/g), the average per capita consumption of *nfuma* (C*_nfuma_*: g) and the adult and toddlers standard human body weight (bw) of 70 and 12 kg [23], respectively, according to the following formula:(5)$$\mathrm{EI}=\frac{C_{\mathrm{element}}\;\times \;C_{nfuma}}{\mathrm{bw}}$$

The estimated daily intake (EDI: µg/day/kg bw) of Ni, weekly intake (EWI: µg/week/kg bw) of Al and monthly intake (EMI: µg/month/kg bw) of Cd were calculated assuming a C*_nfuma_* of 100 g, 700 g and 3000 g, respectively.

The obtained EI were expressed as % of the toxicological guidance levels for exposure assessment, namely the tolerable daily intake (TDI) [24], the provisional tolerable weekly intake (PTWI) and provisional tolerable monthly intake (PTMI) [25], for Ni, Al, Cd, respectively.

#### 2.2.8. Statistical Analysis

All samples were prepared and analyzed in triplicate. Data were tested for normal distribution of the residuals with Shapiro–Wilk test. The existence of statistically significant differences between means was studied using one-way analysis of variance (ANOVA), when the normal distribution of residuals was confirmed. Welch correction was applied when the homogeneity of variances was not verified. Whenever statistical significance was found, Tukey’s or Tamhane’s T2 post hoc tests were applied to compare the means, depending, respectively, on equal variance or not. All these statistical analyses were conducted with the XLSTAT for Windows version 2014.5 (Addinsoft, Paris, France) at the 0.05 significance level. 

## 3. Results and Discussion

### 3.1. Proximate Composition, Energy Value and Sugar Profile

The macronutrient composition of *S. madagascariensis* fruit flour (*nfuma*) is presented in Table 1. The main component was available carbohydrates (49.7–54.9%), followed by fat (26.3–27.8%), total fiber (5.8–10.8%), ash (4.8–6.0%), moisture (~4%) and protein (~3%). *Nfuma* provides an energy value of approximately 475–490 kcal/100 g of flour. 

The moisture content of *nfuma* ranged from 4.4 to 4.6%, with no significant differences between the different origins (districts). *Nfuma* showed a low moisture content when compared to cereal flours such as maize and wheat (~13.4) [26]. Compared to commercial fruit flours, it presented a higher content than coconut flour (3.8%), but lower than grape (5%) and other fruit flours (up to 12.2%). The low moisture content of *nfuma*, together with proper packaging, minimizes the risk of microbiological contamination as well as product deterioration during storage, while also improving shelf life.

Protein (3.0–3.5%) was the macronutrient found in the lowest content, similar to flours from other fruits such as green bananas (3.4%) [1] and other common fruits, as described by Carli et al. [27] for 10 different commercial fruit flours (1.59–6.59%). Regarding *Strychnos* spp, Ngadze et al. [9] reported a low protein content for monkey orange, especially for *S. innocua* (0.3–11.5%). [28] also found low protein contents in indigenous fruits (1.3–3.7% dw), reporting 3.3% for *S. spinosa*.

*Nfuma* has a high-fat content (26.3–27.8%) (Table 1), with the flour originating in Manhiça having the highest value (*p* < 0.05), with no significant differences between flour from other communities. The fat content of *nfuma* is higher than the values reported by Ngadze et al. [9] for all *Strychnos* spp. (0.3–20%), except for *S. spinosa,* where a content of 31.2% dw was reported [29], which was considered an outlier [9]. This flour has an unusual high fat content, even when compared with flours from fat-rich fruits, such as coconut and nuts, because those are by-products of the oil/vegetable milk extraction process [30].

The fixed residue (ash) in *nfuma* ranged from 4.8 to 6.0%. The values obtained are similar to those described for *S. innocua* (4.7%) [31]. Compared to commercial fruit flours studied by Carli et al. [27], passion fruits, papaya and açai presented values (4.9–6.5%) in the same range of *nfuma*.

The most abundant macronutrient of *nfuma* was carbohydrate (~60%). Available carbohydrates varied from 49.7 to 54.9%, with the lowest content found in Chókwè and the highest in Marracuene flour. This value is lower than >80% of total carbohydrates found in other flours, such as cereal and cassava flours [26] or unripe banana and sweet potato flours [32]. In relation to *Strychnos* Spp, lower values of total carbohydrates were described in the Ngadze et al. review [9]; however, Kalenga Saka and Msonthi [29] described a total carbohydrate value of 61% dw in *S. innocua*. Regarding the free sugars, which together represent ~10% of flour mass, fructose and glucose are the ones present in higher levels, ranging between 3.3–4.3% and 3.7–5.0%, respectively, while sucrose represents only 1.4–1.9% of the flour. No significant differences (*p* < 0.05) were observed between sugars profiles of flours from different communities. The total fiber content of *nfuma* varies significantly between communities, with Marracuene flour having the lowest content of total fiber (5.8%) and Chókwè flour the highest one (10.8%). In fact, the highest soluble fiber content in Chókwè (3.9% vs. 0.6–1.4%) is responsible for the higher total fiber content of these flour, since the insoluble fiber value is similar in all flours (5.2–6.9%). Ngadze et al. [9], reported lower values of total carbohydrates for *Strychnos* spp. (*S. innocua* and *S. spinosa*, *S. cocculoides* and *S. pungens*), however Kalenga Saka and Msonth [29] and Lockett et al. [33] described a total carbohydrate value of ~60% for *S. innocua* and *S. spinosa*, respectively. Regarding the sugar profile, the higher ratio of monosaccharides/sucrose agrees with [34], wherein most of the indigenous South African fruits were monosaccharides dominant. This pattern was also observed in several dried fruits [35]. 

Given the total fiber content (ranging between 6 and 11%), *nfuma* can be considered as a food “high in fiber”, according to Regulation (EC) No. 1924/2006 [36], a claim which “may only be made where the product contains at least 6 g of fiber per 100 g”. Compared with cassava four (1.6%) and corn flour (2.6%) [18], *nfuma* stands out for its high fiber content. In the study by Carli et al. [27], only coconut (9.4%) and orange (7.6%) flours had similarly high fiber contents. The other fruit flours studied had fiber contents between 2.0–4.9%. Ngadze et al. [9] describe a fiber content of 2.5–22.2% dry weight basis (dw) for other *Strychnos* spp, with the mean fiber content of *S. innocua* (9.4%) the most similar to *nfuma*, the *S. madagascariensis* flour. 

### 3.2. Fatty Acids Composition and Atherogenic and Thrombogenic Indices

*Nfuma* has a high percentage of monounsaturated fatty acids-MUFA (~17 g/100 g of flour; ~65% of total fatty acids), followed by saturated fatty acids-SFA (~7 g/100 g; ~25% of total fatty acids) and polyunsaturated fatty acids-PUFA (~2.5 g/100 g; ~9% of total fatty acids) (Table 2). In relation to the origin of the flour, significant differences (*p* < 0.05) were observed for the main fatty acids, oleic and palmitic acids, being the flour from Manhiça significantly higher in both fatty acids, which is in line with its higher fat content.

The fatty acid profile of *nfuma* is similar to that of high-fat fruits and their oils, namely olives, avocados and nuts [37,38], but with a slightly higher saturated content. Although *nfuma* has a higher fat content than cereal flour, it presents high amounts of oleic acid (~16 g/100 g of flour), some linoleic (1.8 g/100 g) and alpha-linolenic (0.5 g/100 g) acids and low amounts of SFA, which means it can play a protective role in health. The SFA fraction was mainly palmitic acid (~5 g/100 g), known for its controversial association with detrimental health effects; however, an optimal intake of palmitic acid, in an adequate ratio to unsaturated fatty acids may be crucial to maintain membrane phospholipids balance [39]. 

Despite the differences in the main fatty acids, regarding the health-related lipid indices of *nfuma*, the atherogenic index (AI) and thrombogenic index (TI) were 0.28 and 0.60, respectively, and did not differ between flour origins. Lower AI and TI values (close to zero) translate into lower atherogenic and thrombogenic potential. Thus, the consumption of *nfuma* can contribute to the prevention of cardiovascular diseases, since this flour, like olive oil (AI = 0.14; TI = 0.32), has AI and TI values below 1 [16]. Higher indices were reported for coconut (AI = 13.63; TI = 6.18) and palm (AI = 2.03; TI = 2.07) lipids, which are mainly characterized by SFA and are associated with cardiovascular diseases [16,40].

### 3.3. Vitamin E and Carotenoids

Unlike cereal flours, the high-fat content of *nfuma* is also responsible for its high content of liposoluble bioactive compounds, namely vitamin E and β-carotene (provitamin A activity) (Table 3). 

Vitamin E and β-carotene contents ranged between 6.73–7.97 and 2.19–2.64 mg/100 g, respectively. Although no statistically significant differences were observed between communities, Chókwè and Manhiça presented the highest content of vitamin E and β-carotene, respectively. *Nfuma* has a higher β-carotene content than yellow sweet potato flour (0.6 mg/100 g) which is commonly used by local communities in Mozambique to make bread, cakes and porridges [41]. Regarding vitamin E, *nfuma* was shown to have a significant amount, similar to peanuts (9.9 mg/100 g) and some vegetable oils, such as palm oil (9.5 mg/100 g) [26].

### 3.4. Amino Acids and Protein Nutritional Quality

The amino acid (AA) composition (mg/g protein) of *nfuma* from the different districts is shown in Table 4. Of the 18 amino acids analyzed, 17 were identified in the flour. Arg was the amino acid with the highest amount (148–156 mg/g protein), followed by Asp (106–121 mg/g), Glu (92–103 mg/g) and Ser (93–97 mg/g), Val (73–76 mg/g) and Leu (73–75 mg/g), Thr (58–61 mg/g) and Phe (59–62 mg/g) and Cys (50–56 mg/g), with significant differences between communities (*p* < 0.05) for Arg, Asp and Ser. Protein from Marracuene flour presented the higher Asp and Ser content, and lower of Arg. The lowest AA amounts were found for Met, Trp and Lys (up to 17 mg/g protein), being significantly different between communities, and Ala was not detected. Lys and Met were significantly higher in protein from Chicualacuala flour (16.9 mg/g protein) and Chókwè (15.6 mg/g protein), respectively, and Trp was significantly lower in protein from Marracuene flour (10.3 mg/g protein). Studies on the AA composition of indigenous fruits is scarce in the literature [42], and no information is available for fruits or products of *Strychnos* spp. [9]. Arg, the main AA in *nfuma*, was also found as the main AA of *Dovyalis longispina* and is generally abundant in other indigenous fruits [42]. Glu and Asp, the second and third most abundant AA in *nfuma*, have been described in relatively high amounts in other indigenous fruits [42]. These three AA have been described as major amino acids in nut seeds [38].

Met and Trp were the least abundant AA in *nfuma* (up to 15.6 mg/g protein), which is a common finding for proteins of plant origin [19]. Met was also found to be the least abundant AA by Sibiya et al. [42] in almost all indigenous fruits; however, different results were found for Lys, one of the least abundant AA in *nfuma* (up to 16.9 mg/g protein), but with relatively high or intermediate levels in some indigenous fruits [42]. Lys exhibits significant thermal instability even at low temperatures [19]; therefore, some losses may have occurred during the drying and roasting processes and its concentration may be higher in *S. madagascariensis* fresh fruits. Although in low amounts, *nfuma* presents all nine essential AA, which is in agreement with the findings of Sibiya et al. [42], who found eight essential AA (Trp was not evaluated) in 14 indigenous fruits. 

The nutritional quality of the protein of *nfuma*, expressed as essential amino acid scores (EAAS) [20], is presented in Figure 2.

When all nine individual scores are greater than or equal to 1, the protein is considered complete. Despite having all essential AA, according to the results obtained, the protein of *nfuma* is incomplete, presenting Lys as the limiting AA (AAS < 1), which is in agreement with the results available for the protein of other flours, namely those derived from cereals, e.g., wheat and corn [19], or nut seeds [38]. Although the protein of *nfuma* is incomplete, it can balance other amino acid deficiencies. *Nfuma* is a relatively important source of Met+Cys, Phe+Tyr and Thr (AAS ≥ 1). To achieve the recommended daily allowances of all EAA, local communities should combine *nfuma* with other protein sources, namely legumes such as beans, as they are a rich source of Lys and a poor source of Met [19]. Beans are mainly grown in rural areas of Mozambique and can help to alleviate malnutrition [43]. The EAA with the highest ratio to daily requirements was Phe+Tyr. These AA are precursors of the physiologically active molecules catecholamines, which act as both neurotransmitters and hormones [44]. Although the three most abundant AA in *nfuma* are considered non-essential, it has been shown that Arg, Asp and Glu (Table 4) can act as regulators of key metabolic pathways, leading to a new concept of functional AA [45]. 

### 3.5. Mineral Elements

Table 5 presents the results of mineral content in *nfuma* from different origins (districts). Of the 25 elements analyzed, the content of eight of them (Li, Be, As, Se, Cs, Pb, Tl and Bi) was below the limits of detection. The most abundant macromineral was K (ranging from approximately 1200 to 1700 mg/100 g) and the less abundant was Na (4.0–6.6 mg/100 g). 

This trend was also observed in other fruit flours [1,27], dried fruits [46] and indigenous fruits [28,47]. Brito et al. [1] found similar values for green banana flour, with mean value of 1100, 88, and 45 mg/100 g for K, Mg and Ca, respectively. Commercial fruit flours (up to 952 mg/100 g) [27] and dried fruit products (up to 1162 mg/100 g), have been described as essential sources of K [46]; however, *nfuma* presents even higher values. Among the 14 wild fruits native to southern African studied by Sibiya et al. [47], higher K levels were found in *Carissa macrocarpa* and *Syzygium cordatum* (1312.3 and 1427.1 mg/100 g dw, respectively). Regarding *Strychnos* spp, our results for *S. madagascariensis* flour are in the range of those observed by Kalenga Saka and Msonthi [29] and Amarteifio and Mosase [28] for *S. spinosa* fruit (1968 and 1370 mg/100 g dw, respectively) for K, and higher for Mg (43 and 49 mg/100 g dw, respectively).

For essential macrominerals, statistically significant differences were observed between the content of K, Mg and Na in the flours of the different communities (Marracuene, Chókwè, Chicualacuala and Manhiça). A high K and Mg content was observed for Marracuene flour (1654 ± 102 and 85.6 ± 2.8 mg/100 g, respectively) and a high Na content was observed for Chicualacuala flour (6.6 ± 0.1 mg/100 g).

Regarding essential trace elements, the following trend was observed: Mn (~4000 µg/100 g) > Fe > Zn > Cu > Cr > Co (6.8–7.6 µg/100 g). Significant differences (*p* < 0.05) were observed only for Cu (higher Cu content in Marracuene flour compared to Chicualacuala flour). *Nfuma* can be considered an interesting source of Mn when compared to dried fruits (~300 µg/100 g: apricot, dates, peach, poir, runes, raisins) [46]. Of the commercial fruit flours studied by Carli et al. [27], only coconut showed a Mn concentration similar to *nfuma*. Hassan et al. [48] found a Mn content of 2500 µg/100 g dw for *S. innocua* fruit. For Fe (~1600 µg/100 g) and Zn (~250 µg/100 g) in *nfuma*, lower levels were found when compared to cashew flours (~5000 and 4000 µg/100 g, respectively) [49]. Carli et al. (2017) reported higher Fe values in 8 out 10 fruit flours (~3000–11,000 µg/100 g: plum, coconut, orange, papaya, apple, passion fruit, green banana flours, in ascending order) and all flours had higher Zn values (500 to 4000 µg/100 g) than *nfuma*. Amarteifio and Mosase [28] reported a similar Zn content (220 µg/100 g dw) for *S. spinosa* fruit, and a lower value for Fe (~110 µg/100 g dw), proposing the supplementation of these fruits to meet Fe requirements. Interestingly, several studies report the ability of *Strychnos* spp. to improve nutrition based on their high Fe and Zn contents [9,10,50], despite the wide variation between and within the *Strychnos* spp. (other than *S. madagascariensis*), as reviewed by Ngadze et al. [9]. The Cu content (200 µg/100 g) of *nfuma* was in agreement with the data for some fruit flours, as reported by Carli et al. [27] (~170–210 µg/100 g for açai, orange and lemon flours) and Brito et al. [1] (up to 300 µg/100 g for apple and green banana). Regarding Cr, *nfuma* has contents (~58 µg/100 g) higher than those reported by Brito et al. [1] for apple and green banana flours (up to 20 µg/100 g). The minor essential element found in *nfuma* was Co (~7 µg/100 g), a constituent of vitamin B12. *Nfuma* presents higher Co levels than grain flours (up to 1.2 µg/100 g) [51] and within the range of dried sweet cherries (~0.6 to 14 µg/100 g; mean value of 3 µg/100 g dw) [52]. For *S. innocua*, Hassan et al. [48] reported quite different Co levels compared to *nfuma*, reaching 1200 µg/100 g dw. 

Some non-essential elements were also quantified in *nfuma*. Among those, Al was the most abundant (~2600–3300 µg/100 g) and Cd the least (~2.1–2.4 µg/100 g). Statistically significant (*p* < 0.05) were observed for Al, Rb and Ba, with a higher Rb content and a lower Ba content in the Marracuene flour. For Al, a higher content was observed for Manhiça flour compared to Marracuene flour (3331 ± 26 vs. 2631 ± 210 µg/100 g). Aluminum content can vary significantly in food, depending on the food composition itself as well as on “external” factors (e.g., soil contamination, culinary practices). Our results are in close agreement with Brito et al. (2017), who studied two fruit flours (apple and green banana) and found Al levels ranging from 190 to 4900 µg/100 g.

### 3.6. Nutrient Adequacy of Nfuma 

The estimated daily intake of nutrients and energy, expressed as % of DRV, was calculated assuming an average per capita consumption of *nfuma* of 100 g per day and is presented in Table 6. For macronutrients, except protein and total dietary fiber, the EDI was calculated based on the energy value. Thus, macronutrients DRVs depend on each individual’s energy needs. Overall, consumption of 100 g of *nfuma* contributes to 15–27% of daily energy needs, depending on the physical activity level. EDI values show that *nfuma* is an important source of fiber and lipids, namely alpha-linolenic acid, contributing to 30%, 22–69% and 27–48% of DRVs, respectively. Regarding liposoluble vitamins, *nfuma* provides 55–63% and 56–66% of vitamins A and E, respectively. In Mozambique, 69% of children under 5 years of age are deficient in vitamin A (Amaro, 2019; World Health, 2006). Considering a daily consumption of 50 g of *nfuma*, children aged 1–3 and 4–6 years can obtain 82 and 68% (Table 7), respectively, of vitamin A EDIs [22]. Thus, the consumption of *nfuma* by children in Mozambique may alleviate vitamin A deficiency due to its high β-carotene content. As mentioned above, flour is used during times of food shortage as a supplement to staple foods, such as maize and cassava flour. Maize flour is often boiled in water to make a maize-meal porridge that is consumed for breakfast by communities in sub-Saharan Africa [9]. According to the FAO Food Balance Sheets, Mozambican communities consume 192 g of maize and 285 g of cassava per day [53]. Maize and cassava flour have a higher carbohydrate content (75% and 85%, respectively), while *nfuma* has more fiber, fat and liposoluble vitamins [26]. On the other hand, *nfuma* has a low protein content, providing only 6% (male) and 7% (female) of DRVs. Therefore, as noted above, *nfuma* must be combined with other protein sources to achieve DRVs.

Considering the EDI of essential elements (expressed as % of DRV), *nfuma* contributes significantly to the daily intake of Mg and K, representing 22–26% and 40% of DRVs, respectively. In addition, daily consumption of 100 g of *nfuma* is sufficient to meet Mn requirements, since it provides more than 100% of Mn DRV. *Nfuma* has a higher content of specific minerals, especially K and Mg, compared to common staple flours in Mozambique, such as maize (120 mg/100 g of K and 46 mg/100 g of Mg) and cassava flours (20 mg/100 of K and 2 mg/100 g Mg) [26]. On the other hand, 100 g of *nfuma* provides only 10% (female) to 15% (male) of Fe DRV, although it contains twice the Fe content of maize flour (800 µg/100 g). For children (1–6 years), 50 g of *nfuma* provides 12% of Fe DRV (Table 7).

Consequently, the combination of maize flour with *nfuma* in porridges, together with consumption of *nfuma* as a daytime snack, as reported by people from Mozambican communities, may increase Fe intake, which is of great importance given the high prevalence (64%) of anemia in children in Mozambique [54]. However, it is still necessary to study the bioaccessibility and bioavailability of *nfuma* minerals to define its real potential in human nutrition.

### 3.7. Exposure Assessment to Non-Essential Trace Elements

Considering the non-essential trace elements for which a tolerable intake is established, the amount of Ni, Al and Cd in *nfuma* contributes to 52%, 15% and 9% of the corresponding TDI, PTWI and PTMI, respectively, for adults (Table 8). Thus, the consumption of *nfuma* (100 g) alone is not likely to be considered a relevant source of Al and Cd. The same is not true for Ni, since the average content found contributed to ~50% of the TDI established by the European Food Safety Authority (13 µg/kg bw).

When looking at the exposure of toddlers, the consumption of 50 g of *nfuma* can contribute to 151%, 44% and 11% of the TDI, PTWI and PTMI of Ni, Al and Cd, respectively. These results indicate that young age groups can be at high risk of health complications due to Ni exposure. Since the fruits used to prepare the flour had low levels of Al and Ni (data not shown), the selection of appropriate materials and the use of good practices in the preparation of *nfuma* should be evaluated in order to mitigate the presence of these non-essential elements, while maintaining the nutritional advantages of the food discussed above.

## 4. Conclusions

This research aimed to determine the nutritional composition of *nfuma*, a flour from *S. madagascariensis* pulp fruit, prepared by local communities in Mozambique and evaluate its adequacy in terms of nutrient recommendations. This fruit flour stands out for its high fat content, mainly composed by MUFA, delivering vitamin E and carotenes, together with naturally occurring sugars and high fiber content. *Nfuma* is also a good source of Mn and K and, despite being a poor source of Fe, *nfuma* contains twice the Fe content of maize flour. However, its Ni content should be addressed with caution and mitigation strategies are required in order to guarantee its safety. 

Although *nfuma* bioaccessibility evaluation is still needed, its consumption seems to be a promising food-based strategy to alleviate the high prevalence of anemia and vitamin A deficiency in children of Mozambique. Its local use in the “enrichment” of maize-based porridges or as ingredient for pastry and snacks for the development of healthier new food products deserves to be technologically approached for wider valorization.

## Figures and Tables

**Figure 1 foods-11-00616-f001:**
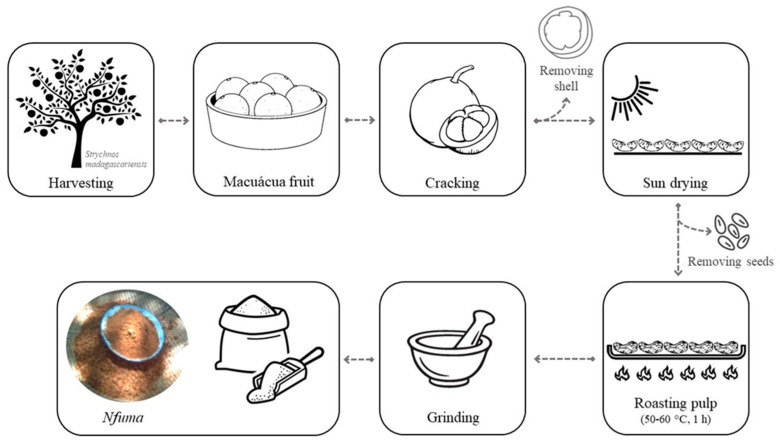
Schematic description of the preparation of *nfuma* by Mozambican communities: from the fruit of *Strychnos madagascariensis* to flour.

**Figure 2 foods-11-00616-f002:**
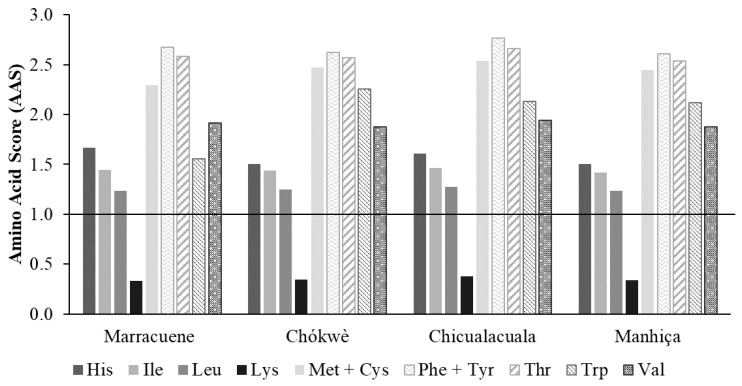
Essential amino acid scores (EAAS) of *S. madagascariensis* fruit flour (*nfuma*) from 4 districts of Mozambique. Adult maintenance patterns are expressed as mg AA/g protein: His. 15; Ile. 30; Leu. 59; Lys. 45; Met + Cys. 27; Phe + Tyr. 38; Thr. 23; Trp. 6.6; Val. 39, according with WHO/FAO/UNU [20].

**Table 1 foods-11-00616-t001:** Proximate composition, energy value, sugar and fiber profiles (%) of *S. madagascariensis* fruit flour (*nfuma*) from four districts (Marracuene, Manhiça, Chókwè and Chicualacuala) in southern Mozambique.

		Marracuene	Chókwè	Chicualacuala	Manhiça	*p* Value
**Moisture** (g)		4.4 ± 0.2	4.6 ± 0.0	4.3 ± 0.1	4.4 ± 0.1	ns
**Protein** (g)		3.1 ± 0.0 ^b^	3.2 ± 0.1 ^b^	3.0 ± 0.0 ^b^	3.5 ± 0.1 ^a^	0.003 **
**Fat** (g)		26.3 ± 0.2 ^b^	27.0 ± 0.0 ^b^	26.9 ± 0.1 ^b^	27.8 ± 0.0 ^a^	0.016 **
**Ash** (g)		5.5 ± 0.1 ^b^	4.8 ± 0.0 ^c^	6.0 ± 0.1 ^a^	5.0 ± 0.1 ^c^	0.020 **
**Total Carbohydrates** ^#^ (g)		60.7 ± 0.1 ^a^	60.5 ± 0.1 ^a,b^	59.8 ± 0.2 ^a,b^	59.3 ± 0.1 ^b^	0.026 **
**Available Carbohydrates** ^#^ (g)		54.9 ± 0.1 ^a^	49.7 ± 0.1 ^d^	52.4 ± 0.0 ^c^	53.1 ± 0.1 ^b^	0.006 **
Sugars						
Fructose (g)		4.0 ± 0.0	3.5 ± 0.1	3.3 ± 0.2	4.3 ± 0.5	ns
Glucose (g)		4.4 ± 0.2	3.7 ± 0.0	3.7 ± 0.1	5.0 ± 1.0	ns
Sucrose (g)		1.4 ± 0.2	1.8 ± 0.4	1.9 ± 0.4	1.8 ± 0.4	ns
**Total dietary fiber** (g)		5.8 ± 0.1 ^c^	10.8 ± 0.0 ^a^	7.4 ± 0.2 ^b^	6.2 ± 0.1 ^c^	0.001 **
Insoluble (g)		5.2 ± 0.1	6.9 ± 0.4	6.0 ± 0.2	5.5 ± 0.2	ns
Soluble (g)		0.6 ± 0.1 ^c^	3.9 ± 0.4 ^a^	1.4 ± 0.01 ^b^	0.6 ± 0.1 ^c^	0.022 **
**Energy**	(kJ)	2005 ± 8 ^a,b^	1982 ± 1 ^b,c^	1995 ± 1 ^a^	2039 ± 3 ^a^	0.012 **
	(kcal)	480 ± 2 ^a,b^	476 ± 0.2 ^b,c^	478 ± 0.2 ^a^	489 ± 0.7 ^a^	0.014 **

Data expressed as mean ± standard deviation (*n* = 3 independent samples × 3 analytical replicates); ns, not significant. Different letters for each district in a row indicate statistically significant differences (*p* < 0.05) between means. *p* Values from one-way ANOVA. Means were compared by Tukey’s since homogeneity of variances was confirmed by Levene’s test (*p* > 0.05). ** *p* Values from one-way Welch ANOVA. Means were compared by Tamhane’s T2 test since homogeneity of variances was not confirmed by Levene’s test (*p* < 0.05). ^#^ Carbohydrate content was determined by difference [Total carbohydrates = 100 − (moisture + protein + fat + ash); Available carbohydrates = Total carbohydrates − Total dietary fiber].

**Table 2 foods-11-00616-t002:** Fatty acids composition (mg/100 g of flour) and atherogenic (AI) and thrombogenic (TI) indices of *S. madagascariensis* fruit flour (*nfuma*) from four districts (Marracuene, Manhiça, Chókwè and Chicualacuala) in southern Mozambique.

Fatty Acid	Marracuene	Chókwè	Chicualacuala	Manhiça		*p* Value
C12:0	3.4 ± 0.2	3.5 ± 0.9	3.8 ± 0.4	3.7 ± 0.5		ns
C14:0	34.1 ± 1.9	35.4 ± 1.0	35.3 ± 1.2	36.1 ± 0.5		ns
C16:0	5229.9 ± 118.3 ^a,b^	5354.5 ± 58.6 ^a^	5359.1 ± 67.2 ^a^	5528.3 ± 52.6 ^b,c^		0.033 *
C16:1-*n*9	421.4 ± 13.5	428.7 ± 3.1	428.5 ± 8.9	439.1 ± 1.8		ns
C18:0	1198.7 ± 13.2	1201.6 ± 23.3	1213.6 ± 15.8	1258.0 ± 24.5		ns
C18:1-*n*9	16,464.1 ± 298.7 ^a,b^	16,905.2 ± 3.3 ^c^	16,812.7 ± 147.7 ^a,c^	17,388.1 ± 25.0 ^b,c^		<0.001 **
C18:2-*n*6	1791.6 ± 55.4	1832.8 ± 17.6	1839.6 ± 36.2	1884.9 ± 13.0		ns
C18:3-*n*3	457.5 ± 21.1	460.8 ± 9.9	466.0 ± 14.1	472.4 ± 9.6		ns
C20:0	161.0 ± 2.8	165.7 ± 2.5	168.2 ± 5.4	171.8 ± 2.8		ns
C20:1-*n*9	78.1 ± 1.4	82.4 ± 2.7	83.3 ± 3.9	84.8 ± 3.7		ns
SFA	6926 ± 138	7068 ± 67	7084 ± 76	7314 ± 80		-
MUFA	17,013 ± 313	17,469 ± 4	17,390 ± 160	17,965 ± 17		-
PUFA	2304 ± 79	2345 ± 27	2361 ± 51	2408 ± 17		-
**AI** ^#^	0.28 ± 0.00	0.28 ± 0.00	0.28 ± 0.00	0.28 ± 0.00		-
**TI** ^#^	0.60 ± 0.00	0.60 ± 0.00	0.60 ± 0.00	0.60 ± 0.00		-

Data expressed as mean ± standard deviation (*n* = 3 independent samples × 3 analytical replicates); ns, not significant. Different letters for each district in a row show statistically significant differences (*p* < 0.05) between means. * *p* Values from one-way ANOVA. Means were compared by Tukey’s since homogeneity of variances was confirmed by Levene’s test (*p* > 0.05). ** *p* Values from one-way Welch ANOVA. Means were compared by Tamhane’s T2 test since homogeneity of variances was not confirmed by Levene’s test (*p* < 0.05). ^#^ AI and TI were calculated according to Ulbricht and Southgate [16].

**Table 3 foods-11-00616-t003:** Vitamin E and β-carotene contents (mg/100 g) of *S. madagascariensis* fruit flour (*nfuma*) from four districts (Marracuene, Manhiça, Chókwè and Chicualacuala) of southern Mozambique.

	Vitamin E	β-Carotene
Marracuene	6.73 ± 0.74	2.56 ± 0.08
Chókwè	7.97 ± 0.72	2.45 ± 0.34
Chicualacuala	6.88 ± 0.13	2.19 ± 0.11
Manhiça	7.44 ± 0.40	2.64 ± 0.13
*p* value	ns	ns

Data expressed as mean ± standard deviation (*n* = 3 independent samples × 3 analytical replicates); ns, not significant.

**Table 4 foods-11-00616-t004:** Amino acids composition of protein (mg/g protein) from the *S. madagascariensis* fruit flour (*nfuma*) from four districts (Marracuene, Manhiça, Chókwè and Chicualacuala) in southern Mozambique.

Amino Acid	Marracuene	Chókwè	Chicualacuala	Manhiça	*p* Value
Asp	121 ± 1 ^b^	106 ± 3 ^a^	114 ± 2 ^a,b^	116 ± 4 ^b^	0.004
Glu	101 ± 5	103 ± 17	92.0 ± 2.6	103 ± 18	ns
Ser	97.2 ± 0.4 ^b^	93.2 ± 1.5 ^a,b^	96.9 ± 0.5 ^a,b^	92.7 ± 2.2 ^a^	0.020
His	25.0 ± 2.4	22.6 ± 0.3	24.1 ± 0.6	22.6 ± 0.7	ns
Gly	27.3 ± 0.4	27.3 ± 0.4	27.8 ± 0.2	27.7 ± 0.5	ns
Thr	59.5 ± 0.8	59.1 ± 0.7	61.1 ± 1.7	58.4 ± 1.5	ns
Arg	148 ± 1 ^b,c^	153 ± 2 ^a,c^	156 ± 1 ^a^	153 ± 3 ^a,c^	0.041
Ala	nd	nd	nd	nd	
Tyr	42.6 ± 0.3	40.7 ± 0.6	42.7 ± 0.1	40.3 ± 1.5	ns
Val	74.6 ± 0.3	73.1 ± 0.9	75.6 ± 0.3	73.1 ± 2.3	ns
Met	12.0 ± 0.0 ^b^	15.6 ± 0.3 ^a^	11.5 ± 0.3 ^b,c^	10.9 ± 0.2 ^c^	<0.001
Phe	59.0 ± 0.2	59.0 ± 0.8	62.4 ± 0.2	58.9 ± 2.4	ns
Ile	43.3 ± 0.6	43.1 ± 1.1	43.8 ± 0.3	42.6 ± 1.1	ns
Leu	72.8 ± 0.3	73.6 ± 1.2	75 ± 0.3	72.7 ± 2.2	ns
Lys	14.9 ± 0.1 ^b^	15.6 ± 0.2 ^b^	16.9 ± 0.3 ^a^	15.4 ± 0.5 ^b^	0.002
Pro	41.1 ± 1.5 ^a,b^	49 ± 5.2 ^b^	30.1 ± 3.1 ^a^	43.1 ± 3.6 ^b^	0.005
Trp	10.3 ± 0.5 ^b^	14.9 ± 0.3 ^a^	14.0 ± 0.6 ^a^	14.0 ± 0.3 ^a^	<0.001
Cys	49.9 ± 1.6	50.4 ± 4.3	56.4 ± 1.3	55.5 ± 1.0	ns
∑AAA ^1^	102 ± 0	100 ± 1	105 ± 0	99.2 ± 3.9	-
∑SAA ^2^	61.9 ± 0.0	66.8 ± 0.3	68.4 ± 0.3	66.1 ± 0.2	-
∑EAA ^3^	464 ± 3	469 ± 6	484 ± 1	464 ± 12	-

Data expressed as mean ± standard deviation (*n* = 3 independent samples × 3 analytical replicates); nd, below limit of detection (4.06 µM for Ala); ns, not significant. Different letters for each district in a row show statistically significant differences (*p* < 0.05) between means. *p* Values from one-way ANOVA analysis. Means were compared by Tukey’s. since homogeneity of variances was confirmed by Levene’s test (*p* > 0.05). ^1^ Aromatic amino acids: Phe+Tyr; ^2^ sulfur amino acids: Met+Cys; ^3^ Sum of essential amino acids Thr+Val+Met(+Cys)+Ile+Leu+Phe (+Tyr)+His+Lys+Trp used for daily requirements and protein value as suggested by FAO/WHO/UNU [20].

**Table 5 foods-11-00616-t005:** Mineral composition of the *S. madagascariensis* fruit flour (*nfuma*) from four districts (Marracuene, Manhiça, Chókwè and Chicualacuala) in southern Mozambique.

Element	Marracuene	Chókwè	Chicualacuala	Manhiça	*p* Value
Essential macrominerals (mg/100 g)					
Ca	24.7 ± 0.2	31.3 ± 6.3	26.2 ± 0.0	29.9 ± 2.5	ns
Mg	85.6 ± 2.8 ^a^	80.8 ± 3.4 ^a,b^	75.3 ± 4.7 ^a,b^	69.4 ± 4.3 ^b^	0.025 *
K	1654 ± 102 ^a^	1399 ± 71 ^b^	1303 ± 15 ^b^	1204 ± 80 ^b^	0.002 *
Na	4.9 ± 0.2 ^b^	4.0 ± 0.1 ^b^	6.6 ± 0.1 ^a^	4.9 ± 0.8 ^b^	0.002 *
Essential trace elements (µg/100 g)					
Fe	1683 ± 71	1620 ± 56	1476 ± 73	1706 ± 122	ns
Zn	261.4 ± 11.2	216.5 ± 12.5	205 ± 13.4	228.6 ± 20.8	ns
Mn	4017 ± 141	3885 ± 71	3874 ± 21	4098 ± 181	ns
Cu	215.2 ± 1.0 ^a^	200.3 ± 5.9 ^a,b^	193.9 ± 7.7 ^b^	208.5 ± 5.1 ^a,b^	0.021 *
Cr	58.1 ± 1.4	55.2 ± 2.7	57.5 ± 1.9	58.1 ± 0.5	ns
Co	7.6 ± 0.3	7.3 ± 0.2	6.8 ± 0.3	7.6 ± 0.0	ns
Non-essential trace elements (µg/100 g)					
Al	2631 ± 210 ^a^	3042 ± 221 ^a,b^	2995 ± 73 ^a^	3331 ± 26 ^b^	0.026 **
Rb	1100 ± 19 ^a^	980 ± 29 ^b^	968 ± 15 ^b^	968 ± 31 ^b^	0.002 *
Ni	463.8 ± 3.8	482.6 ± 20.5	458.8 ± 12.6	478.0 ± 20.0	ns
Sr	258.2 ± 2.6	240.9 ± 5.5	250.2 ± 14.4	256.2 ± 4.8	ns
Ba	222.3 ± 5.1 ^b^	252.2 ± 8.6 ^a^	249.7 ± 9.5 ^a^	257.0 ± 4.6 ^a^	0.006 *
V	17.9 ± 0.4 ^a^	17.6 ± 0.5 ^a^	17 ± 0.6 ^a^	21.9 ^b^ ± 0.3	0.002 *
Cd	2.4 ± 0.0	2.1 ± 0.2	2.3 ± 0.1	2.1 ± 0.1	ns

Data expressed as mean ± standard deviation (*n* = 3 independent samples × 3 analytical replicates); ns, not significant. Different letters for each district in a column show statistically significant differences (*p* < 0.05) between means. * *p* Values from one-way ANOVA. Means were compared by Tukey’s since homogeneity of variances was confirmed by Levene’s test (*p* > 0.05). ** *p* Values from one-way Welch ANOVA. Means were compared by Tamhane’s T2 test since homogeneity of variances was not confirmed by Levene’s test (*p* < 0.05).

**Table 6 foods-11-00616-t006:** Estimated daily intake (EDI), expressed as % of the dietary reference value (DRV) [22], of energy, macronutrients, vitamins and essential elements for adults considering an average per capita *nfuma* consumption of 100 g/day.

	DRV (AI ^a^/AR ^b^/PRI ^c^/RI ^d^/SAI ^e^)		EDI (% DRV)	
	Male	Female	Male	Female
**Energy (MJ/day)**				
Energy	9.1–13.0 ^b,^*	7.4–10.5 ^b,^*	15–22	19–27
**Macronutrients (g/day)**				
Protein	53.1 ^c,#^	45.6 ^c,#^	6.0	7.0
Fat	48.3–120.8 ^d^	38.9–97.3 ^d^	22–56	28–69
Alpha-linolenic acid	1.2–1.7 ^a^	1.0–1.3 ^a^	27–38	33–48
Linoleic acid	9.7–13.8 ^a^	7.8–11.1 ^a^	13–19	17–24
Carbohydrates	244.5–465.8 ^d^	197.1–375.5 ^d^	11–22	14–27
Total Dietary Fiber	25 ^a^		30	
**Vitamins**				
Vitamin A (µg RE/day)	750 ^c^	650 ^c,§^	55	63
Vitamin E (mg/day)	13 ^a^	11 ^a^	56	66
**Minerals (mg/day)**				
Ca	950 ^c^		3	
Mg	350 ^a^	300 ^a^	22	26
K	3500 ^a^		40	
Na	2000 ^e^		0.3	
Fe	11 ^c^	16 ^c,§^	15	10
Zn	9.4–16.3 ^c,†^	7.5–12.7 ^c,†^	1.4–2.4	1.8–3.0
Mn	3 ^a^		132	
Cu	1.6 ^a^	1.3 ^a^	12.8	15.7

* Average requirement for adults (18–79 years) with a physical activity level (PAL) between 1.4 and 2.0. ^#^ Population reference intake for men and women with a reference body weight of 64 and 55 kg, respectively, based on IMC of 22 kg/m^2^. ^§^ Population reference intake for premenopausal women. ^†^ Population reference intake for adults (≥18 years) with a phytate intake level between 300 and 1200 mg/day. DRV: dietary reference value; ^a–e^ reference value applied: AI ^a^: adequate intake; AR ^b^: average requirement; PRI ^c^: population reference intake; RI ^d^: reference intake; SAI ^e^: safe and adequate intake. EDI of vitamin A was calculated based on the conversion of β-carotene content (expressed as mg/100 g) to retinol equivalent (1 µg RE = 6 µg of β-carotene).

**Table 7 foods-11-00616-t007:** Estimated daily intake (EDI), expressed as % of the dietary reference value (DRV) [22], of vitamin A and Fe for children (1–6 years) considering an average per capita *nfuma* consumption of 100 g/day.

	DRV (PRI)		EDI (% DRV)	
	1–3 Years	4–6 Years	1–3 Years	4–6 Years
**Vitamin A** (µg RE/day)	250	300	82	68
**Fe** (mg/day)	7		12	

DRV: dietary reference value; population reference intake. EDI of vitamin A was calculated based on the conversion of β-carotene content (expressed as mg/100 g) to retinol equivalent (1 µg RE = 6 µg of β-carotene).

**Table 8 foods-11-00616-t008:** Estimated daily (EDI), weekly (EWI) and monthly (EMI) intake, expressed as % of toxicological guidance values of Ni, Al and Cd, considering the consumption of 100 g/day (adults) or 50 g/day (toddlers) of *nfuma*.

Element	Reference Value	Estimated Intake of Non-Essential Elements	
Ni	TDI (µg/day/kg bw)	EDI (% TDI)	
		Toddlers	Adult
	13	151	52
Al	PTWI (µg/week/kg bw)	EWI (% PTWI)	
		Toddlers	Adult
	2000	44	15
Cd	PTMI (µg/month/kg bw)	EMI (% PTMI)	
		Toddlers	Adult
	25	11	4

TDI: tolerable daily intake (EFSA 2019); PTWI: provisional tolerable weekly intake; PTMI: provisional tolerable monthly intake (JEFCA 2021).

## Data Availability

Data is contained within the article.

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
