# Peer review of "Nutritional Characterization of *Strychnos madagascariensis* Fruit Flour Produced by Mozambican Communities and Evaluation of Its Contribution to Nutrient Adequacy"

_foods, 2022, doi:10.3390/foods11040616_

Round 1

Reviewer 1 Report

Line 28-29: please avoid this abbreviation in the abstract (AA), do not use the same word twice in a sentence

Line 33-34: specify the abbreviation DRV in full

Line 37-38: keywords with the sentence letter (monkey orange; fruit flour…….)

Line 80: indicate the average weight of one fruit

Table 1: [19959 ± 1a] – is there a mistake?

Full text: ("nufuma", “nufuma”, nufuma ....) different throughout the text, please specify

I recommend creating the conclusion part that would bring together the main results of the research

Reviewer 2 Report

The manuscript presented an evaluation of the nutritional composition of nfuma, the fruit flour of S.madagscariensis, and its nutrient adequacy. Therefore, the scientific contribution of this paper is at a satisfactory level, and the approach and methodology are appropriate for the investigation, but the authors should consider some suggestions and comments:

The authors should reorganize the manuscript according to the journal instructions for authors, separate results from the discussion, and provide conclusions (optional).

  • Provide a concise and precise description of the experimental results, their interpretation, and the experimental conclusions that can be drawn.
  • Discussion: Authors should discuss the results and how they can be interpreted from previous studies and the working hypotheses. The findings and their implications should be discussed in the broadest context possible, and limitations of the work highlighted. Future research directions may also be mentioned. This section may be combined with Results.
  • Conclusions: This section is not mandatory but can be added to the manuscript if the discussion is unusually long or complex.

In the Introduction, Page 2, line 56, consider adding more details about a “lost fruit”. This will give a better insight into the authors’ choice to investigate this fruit.

The Material and Methods, Page 2, line 81, specify the harvesting months.

Page 2, line 86, was the structure of the obtained fruit flour uniform; after using a pestle and mortar, did the authors use some kind of sieve at the nfuma preparation?

 Page 7, provide comments for the proximate composition (Table 1) as they appear in the table, starting with moisture—comment high content of carbohydrates.

Page 8, Table 2, compare and comment on the fatty acid composition of fruit flour from four districts and what district has the optimal flour.

Page 15, line 528, are there any studies about bioaccessibility and bioavailability of nfuma minerals, or are these future research recommendations? 

At the end of the discussion, write in which direction further research should be done and indicate the potential applications of nfuma.
